# Characterization of PEG-Modified Composite Membranes for Microfluidic Oxygenator Applications

**DOI:** 10.3390/mi16121383

**Published:** 2025-12-06

**Authors:** Nicholas C. Higgins, David G. Blauvelt, Shuvo Roy

**Affiliations:** 1Department of Bioengineering and Therapeutic Sciences, University of California, San Francisco, CA 94143, USA; nicholas.higgins@ucsf.edu; 2Department of Pediatrics, Nemours Children’s Health, Wilmington, DE 19803, USA; david.blauvelt@nemours.org; 3Department of Pediatrics, Sidney Kimmel Medical College at Thomas Jefferson University, Philadelphia, PA 19107, USA

**Keywords:** extracorporeal membrane oxygenator (ECMO), silicon membrane, polydimethylsiloxane (PDMS), polyethylene glycol (PEG), microfluidics, hemocompatibility

## Abstract

Microfluidic oxygenators promise to advance extracorporeal membrane oxygenation (ECMO) devices with enhanced hemodynamics and low prime volume. We are developing a silicon-based membrane oxygenator that will offer improved gas transfer and fluid flow control. Polyethylene glycol (PEG) has been used to improve hemocompatibility by providing excellent resistance to protein adsorption. Here, we characterized a polyethylene glycol surface modification of composite silicon–PDMS membranes to evaluate their effects on microfluidic oxygenator properties. X-ray photoelectron spectroscopy (XPS) and water contact angle goniometry confirmed successful PEG attachment, evidenced by the presence of characteristic C-O bonds and increased hydrophilicity, which was stable for 2 weeks. Oxygen flux tests demonstrated gas transfer rates as high as 89.6 ± 17.9 mL/min/m^2^ and 50.8 ± 11.7 mL/min/m^2^ for unmodified and PEG-coated membranes, respectively. Protein adsorption studies with human serum albumin (HSA) demonstrated a significant reduction in nonspecific protein binding on PEG-coated membranes with values as low as 14 ± 6 μg/cm^2^. These studies expand on the characterization of our engineered oxygenator membranes and provide insight for the development of future surface optimization strategies to enhance hemocompatibility.

## 1. Introduction

Extracorporeal membrane oxygenation (ECMO) is a life support technology used in patients with severe cardiac failure, respiratory failure or, in some cases, both. In a conventional ECMO circuit, blood is pumped outside the body to an oxygenator, where gas exchange occurs before the blood is returned to the patient [1,2]. Despite decades of development and growing clinical use, ECMO remains associated with high morbidity and mortality rates [3]. A key contributor to this risk is the oxygenator, the component responsible for gas exchange, which is limited by clot formation and inflammatory responses due to blood–material interactions. These complications increase the risk of thrombotic events such as stroke, organ or limb ischemia, and pulmonary embolism [4,5]. To prevent clotting, patients are administered high doses of systemic anticoagulants, such as unfractionated heparin, which in turn raises the risk of bleeding, including intracranial hemorrhage [6,7,8]. This balance between clotting and bleeding emphasizes the need for the development of oxygenators with improved hemocompatibility.

To address the challenges associated with conventional ECMO, there is growing interest in microfluidic oxygenators that incorporate biomimetic blood flow paths and feature lower priming volumes [9,10,11]. We are developing a microfluidic oxygenator that will utilize composite membranes consisting of a patterned microporous semiconductor silicon backbone bonded to a 5 μm thick gas-permeable polydimethylsiloxane (PDMS) layer [12,13]. The flat, rigid architecture of the patterned silicon chip enables a defined geometry of short and wide channels that ensure precise control over blood flow, reducing stasis and optimizing both pressure drop and wall shear stress. Compared to commercial hollow fiber membranes, this flat-sheet design improves hemodynamics and offers higher oxygen transfer per unit surface area, thus reducing contact time between blood and foreign material. However, the use of wide PDMS channels for improved hemodynamics is not enough to overcome hemocompatibility issues. Therefore, surface modification strategies to improve hemocompatibility while ensuring adequate gas transfer will be critical to the successful development of a functional microfluidic oxygenator.

Polyethylene glycol (PEG) is a widely used hydrophilic polymer that reduces protein adsorption through steric repulsion and increased hydrophilicity and has been applied to improve hemocompatibility in a variety of biomedical devices [14,15]. The compatibility of PEG with microfluidic fabrication techniques and relatively low cost make it a practical candidate for surface modification in PDMS microfluidics and early-stage oxygenator development [16]. PEG has previously been applied to flexible PDMS microfluidics and thin film composites such as polyamide [17,18]. However, the application of PEG to thin film (<10 μm) PDMS has yet to be studied. In this study, we evaluate the effects of PEG coating on surface characteristics and functional properties relevant to oxygenator membrane performance. Specifically, we assess surface hydrophilicity, oxygen flux, and protein adsorption on PEG-coated thin PDMS membranes. These studies provide foundational insight into the role of surface coatings with our composite membrane performance and guide future design of silicon membrane oxygenators.

## 2. Materials and Methods

### 2.1. PDMS Membrane Fabrication

A sacrificial method was used to fabricate the 5 µm surface PDMS layer, in which a temporary PDMS support and PVA release layer enabled transfer of the 5 µm film PDMS onto the silicon substrate [19]. First, a handle layer of degassed 10:1 *w*/*w* (base–curing agent) polydimethylsiloxane (PDMS) (Sylgard 184, Dow Corning, Midland, MI, USA) was prepared by spin-coating onto a 100 mm diameter silicon wafer at 500 rotations-per-minute for 20 s to a thickness of approximately 400 μm, and subsequently cured at 80 °C. A 5% *w*/*w* polyvinyl alcohol (PVA; Sigma-Aldrich, St. Louis, MO, USA) layer was then spin-coated to facilitate subsequent release of the PDMS membrane. This was done by first plasma-treating the PDMS surface in an oxygen plasma cleaner (Expanded Plasma Cleaner, Harrick, NY, USA), followed by spin-coating the PVA solution at 1000 rotations per minute for 30 s and curing at 60 °C. To create the 5 μm PDMS layer, a 1:1 mixture of PDMS and hexane (Sigma-Aldrich, St. Louis, MO, USA) was spin-coated onto the cured PVA layer at 5000 rotations-per-minute for 5 min to a highly uniform and controlled thickness of 5 μm, and subsequently cured. This process resulted in a three-layer stack: handle PDMS, PVA, and the 5 μm thick PDMS membrane. All fabrication steps were performed in a Class 10,000 cleanroom environment to minimize contamination from dust and particulates.

To fabricate the silicon portion of the composite membrane, 150 mm diameter SOI (silicon-on-insulator) wafers (MEMS Engineering & Material, Inc., Sunnyvale, CA, USA) with a 100 μm device layer, 1 μm buried oxide, and 299 μm handle layer (total 400 μm) were patterned with 10 μm × 50 μm pores in SPR-220 photoresist and etched to the oxide using DRIE. The handle side was patterned with 1 mm × 6 mm × 299 μm windows. Wafers were diced and wet-etched with 49% HF to remove the oxide layer, connecting pores to windows and completing the silicon membrane. These membranes were used for oxygen flux testing.

The PDMS stack was cut to match the dimensions of the diced silicon chip. Both the PDMS stack and cleaned silicon membranes were oxygen plasma-treated, and subsequently brought into contact, and cured on a hot plate. Plasma treatment was performed with pure oxygen for 30 s at 30 Watts. After bonding, excess PDMS stacks were trimmed from the edges of the specimens. The samples were then placed in a warm water bath to dissolve the PVA layer, releasing the handle PDMS and leaving the final composite silicon-PDMS membrane (Figure 1A,B).

To reduce time and cost associated with creating patterned silicon, solid silicon chips were used instead for characterization tests that did not involve gas transfer: XPS, water contact angle, and protein adsorption. To serve as substrates for the 5 um-thick PDMS membranes, silicon wafers (MEMS Engineering & Material Inc., Sunnyvale, CA, USA) were diced into 10 mm × 10 mm × 0.4 mm chips. These solid silicon chips underwent a multi-step cleaning process, including organic solvent washes and piranha solution treatment, to prepare the surface for bonding to the PDMS stack (Figure 1C).

Standard PDMS samples (1 cm × 1 cm × 400 µm) were prepared for direct comparison to previous studies using a modified in-house protocol [19]. Sylgard 184 base and curing agent were mixed at a 10:1 *w*/*w* and then degassed in a vacuum chamber for 15 min. A 3 mL bolus of the mixture was deposited on a 100 mm diameter silicon wafer and spin-coated to achieve a thickness of 400 μm, then cured in an oven at 80 °C for 2 h, followed by sizing into test samples.

### 2.2. PEG Coating Procedure

PEG coatings were applied using a modified protocol developed in our laboratory [20]. Composite membranes were plasma-treated in an oxygen plasma oven for 30 s to activate the surface. A 0.5% *v*/*v* PEG solution was prepared by diluting 750 μL of 3-[methoxy(polyethyleneoxy)6-9] propyltrimethoxysilane (Gelest, Morrisville, PA, USA) in 150 mL of isopropyl alcohol (IPA). The solution was stirred and heated to 80 °C. Immediately after plasma treatment, the membranes were transferred to the heated PEG-IPA solution and incubated for 2 h to allow surface bonding. After coating, the membranes were washed thoroughly with IPA and deionized water to remove unbound PEG. These samples were then stored in ambient air or deionized (DI) water for initial surface characterization.

### 2.3. X-Ray Photoelectron Spectroscopy (XPS) Analysis

XPS was performed on the unmodified and PEG-coated membranes as well as standard PDMS samples using a Physical Electronics 5600/5800 system (Physical Electronics, Inc., Chanhassen, MN, USA) at the UC Berkeley Biomolecular Nanotechnology Center. Square (1 cm × 1 cm) samples were prepared and mounted on standard sample holders using double-sided conductive tape. All measurements were performed under ultra-high vacuum (<10^−8^ Torr). Survey scans were acquired over the 0–1100 eV binding energy range with a pass energy of 187.85 eV to determine elemental composition. High-resolution scans were collected for the C1s region using a pass energy of 11.75 eV to enable chemical state analysis.

### 2.4. Contact Angle Goniometry

Changes in surface hydrophilicity were assessed using sessile drop contact angle measurements. All samples were analyzed with an Attension Theta Lite goniometer (Biolin Scientific, Gothenburg, Sweden). A 3.5 μL water droplet was placed at the center of each coated membrane under ambient conditions, and contact angles were recorded at 0.1 s intervals over a 10 s period. Single measurements were taken for each membrane daily for 14 days to evaluate hydrophilicity changes over time. Data analysis was performed using OneAttension software v4.6.6. Some PEG-coated membranes were stored in water, and these samples were dried with house air then placed on a 50 °C hot plate to evaporate any excess water prior to contact angle measurement.

### 2.5. Oxygen Flux Measurements

Oxygen flux through the membranes was measured using a custom-built water flow circuit (*n* = 5 membranes per group). The setup consisted of silicone tubing (Masterflex, Avantor, Radnor, PA, USA), sparged water source, a 3D printed membrane housing, a Cole Parmer roller pump (Masterflex L/S Series, Cole-Parmer, Vernon Hills, IL, USA), and a FOSPOR oxygen measurement probe (NeoFoxGT, Ocean Optics, Orlando, FL, USA) (Figure 2) [13].

Flux was calculated based on the partial pressure of oxygen in the water exiting the membrane housing. Water was first deoxygenated by sparging with nitrogen gas down to a partial pressure of oxygen below 100 mmHg, before being circulated through the system at flow rates of 10 mL/min and 20 mL/min. Pressurized oxygen at 1 PSI and 2.5 PSI was introduced to the gas side of the membrane housing, measured with a pressure sensor (DPI 104, Druck, Baker Hughes, Houston, TX, USA). The dissolved oxygen concentration was measured before and after gas exposure. Henry’s Law (C = kP) was used to calculate oxygen flux where C represents concentration of dissolved oxygen in water, k is a temperature-dependent proportionality constant, and P is the partial pressure of gas.

### 2.6. Protein Adsorption Characterization

To assess protein adsorption, composite membranes were allowed to rest for 1 week post-PEG coating to account for hydrophobic recovery seen in water contact angle studies. Unmodified membranes were also rested for 1 week post-production for direct comparison.

Samples of composite membranes and standard PDMS were placed in a 12-well non-tissue culture-treated plate, and prior to protein exposure, they were pre-conditioned with 2 mL of 0.9% saline and incubated at 37 °C for 1.5 h. Human serum albumin (HSA; Millipore Sigma, Burlington, MA, USA), chosen as a model blood-based protein for its non-specific binding to foreign surfaces, was reconstituted to 1 mg/mL and 2 mg/mL in 0.9% saline and pre-warmed in an incubator at 37 °C. Each sample well was then incubated with 2 mL of the HSA solution for 1.5 h at 37 °C on a 20 RPM orbital shaker [21,22]. After incubation, the protein solution was transferred to LoBind tubes (Eppendorf, Hamburg, Germany) for analysis.

Protein concentration was quantified using the Bicinchoninic Acid (BCA) assay (Thermo Fisher Scientific, Waltham, MA, USA), following the manufacturer’s protocol [23]. Each sample was pipetted into a clear-bottom 96-well plate (Corning Costar, Corning, NY, USA) with 3 replicates per condition, including standard curve and test samples. Absorbance was measured at 562 nm, and results were averaged and normalized to the blank well.

### 2.7. Statistics

A minimum of six samples were collected for water contact angle measurements, and at least five samples were used for protein adsorption and oxygen flux testing. Statistical analyses were performed using Microsoft Excel (Redmond, WA, USA) and GraphPad Prism v10.6.1 (San Diego, CA, USA). We used ANOVA tests to evaluate differences with significance threshold set at α = 0.05. Error bars represent standard deviation unless otherwise noted.

## 3. Results

### 3.1. XPS

XPS was used to analyze changes in the surface chemistry of the PEG-coated membranes (Figure 3). Survey scans of standard PDMS, unmodified membranes, and PEG-coated membranes all showed the expected C1s peak around ~285 eV. When comparing the standard PDMS samples to the unmodified and PEG-coated membranes, a pronounced increase in C1s (285 eV) and O1s (528 eV) peaks, along with a decrease in the Si 2p peak (100 eV), was observed in the survey spectrum. This is consistent with the introduction of carbon and oxygen rich PEG chains, confirming the presence of a PEG layer on the PEG-coated membrane surface [24,25]. High-resolution C1s scans showed the characteristic C-Si peak at 284.4 eV and the emergence of a distinct C–O peak near 286.2 eV in the PEG-coated membranes further supporting the presence of PEG on the modified surface. Looking at ambient and water storage, we saw the same distinct C-O signal with a slightly more noise in the water storage. This could be attributed to excess moisture trapped form the hydrophilic PEG molecules or lower acquisition time. A minor C-O signal was also detected in the unmodified membranes, likely attributed to plasma treatment effects, but minimized upon subsequent hydrophobic recovery [26]. The enhanced C–O signal in the PEG-coated samples confirms that the spin-coating and plasma treatment method can lead to successful surface functionalization of the thin-film PDMS.

### 3.2. Contact Angles

Figure 4 presents the average water contact angle (WCA) measurements for unmodified membranes, PEG-coated membranes (stored under ambient and water conditions), and standard PDMS samples for a standard control over a 14-day period (*n* = 6 per group). On day 0, PEG-coated membranes exhibited a substantial reduction in WCA compared to both untreated groups. The initial WCA for ambient PEG coated membranes was 22° ± 17 (mean ± standard deviation), significantly lower than that of standard PDMS (115° ± 2) and unmodified membranes (90° ± 6) (*p* < 0.01, repeat measures ANOVA for multiple time points & multiple comparisons with Tukey’s correction for each time point). This reduction highlights the immediate impact of PEGylation in enhancing surface hydrophilicity. Over time, partial hydrophobic recovery was observed in the PEG-coated membranes. In ambient-stored PEG-coated membranes, the WCA rose steadily from day 0 through day 4, reaching 58° ± 9 before WCA became stable. Water-stored PEG-coated membranes experienced a more rapid increase, with recovery peaking by day 2, 55° ± 4 before stabilizing. Both PEG-treated groups maintained lower WCAs than untreated PDMS by day 14. Both treatment groups remained significantly lower compared to both untreated groups (*p* < 0.01). Statistical significance is seen between ambient storage and water storage starting at day 13 (*p* < 0.01), indicating that the WCA over time is dependent on storage conditions. The data suggests that storage in water will extend shelf-life of the PEG-coated membranes.

### 3.3. Oxygen Fluxes

Composite membranes were tested at two water flow rates (10 and 20 mL/min) and two oxygen gas pressures (1 and 2.5 PSI) (Figure 5). For unmodified membranes, oxygen flux at 10 mL/min was 0.011 ± 0.002 mL/min at 1 PSI and 0.012 ± 0.003 mL/min at 2.5 PSI (mean ± standard deviation). At 20 mL/min, flux increased to 0.018 ± 0.001 mL/min at 1 PSI and 0.022 ± 0.002 mL/min at 2.5 PSI. By contrast, PEG-coated membranes showed 0.006 ± 0.001 mL/min oxygen flux at 1 PSI and 0.007 ± 0.001 mL/min at 2.5 PSI for 10 mL/min flow, and 0.011 ± 0.002 mL/min at 1 PSI and 0.012 ± 0.002 mL/min at 2.5 PSI for 20 mL/min flow. The maximum surface normalized gas transfer was 89.6 ± 17.9 mL/min/m^2^ (20 mL/min flow rate, 2.5 PSI) for unmodified membranes and 50.8 ± 11.7 mL/min/m^2^ (20 mL/min flow rate, 2.5 PSI) for PEG-coated membranes. Statistical analysis with a three-way repeated measures ANOVA, incorporating coating status and repeated measures on sweep gas pressure and fluid flow rate, showed statistical significance between PEG-coated and unmodified membranes (*p* = 0.02). While there is indeed a decrease in oxygen flux with PEG coatings, this effect can be, at least partially, counteracted by operating at increased fluid flow rates.

### 3.4. Protein Adsorption

To evaluate how PEG coatings affect protein fouling on our membranes, we performed human serum albumin (HSA) adsorption studies at two protein concentrations (1 mg/mL and 2 mg/mL) and across two surface areas (1 cm^2^ and 2 cm^2^) (Figure 6). Adsorbed protein was quantified by measuring the remaining protein in solution using a BCA assay. For 1 cm^2^ surface area and 1 mg/mL HSA, PEG-coated membranes adsorbed 14 ± 6 μg/cm^2^ of protein, significantly less than the unmodified membranes (27 ± 10 μg/cm^2^, *p* = 0.002). A two-way ANOVA was used to analyze the data incorporating factors of membrane type (PEG-coated, unmodified membranes, and standard PDMS) and surface area. From this analysis, we found a statistical significance between the materials (*p* < 0.0001). For 2 mg/mL HSA, PEG-coated membranes adsorbed 22 ± 14 μg/cm^2^, again significantly lower than unmodified membranes (46 ± 5 μg/cm^2^, *p* = 0.03). A two-way ANOVA incorporating sample and albumin concentration was used and showed statistical significance between the sample types (*p* < 0.0001). Increasing the surface area to 2 cm^2^ led to a proportional increase in adsorption across all groups; however, PEG-coated membranes consistently exhibited lower adsorption compared to unmodified membranes and standard PDMS samples (*p* < 0.001 for both conditions). As such, PEG is indeed promising as a coating to enhance hemocompatibility.

## 4. Discussion

Developing a membrane oxygenator for ECMO presents several engineering and biological challenges, particularly around hemocompatibility and gas exchange efficiency. Conventional ECMO membranes, typically composed of polypropylene or polymethylpentene (PMP), offer high gas transfer efficiency but can be limited by poor surface biocompatibility and heterogenous flow dynamics [27,28]. Microfluidic oxygenators using PDMS have improved flow control but face tradeoffs in mechanical stability and hemocompatibility [17,19,29]. Our approach combines a patterned and rigid silicon substrate mounted with a thin PDMS layer to enable controlled, short, and wide flow paths that improve gas exchange efficiency without compromising mechanical integrity. In this study, we characterized the effects of PEG surface modification on these composite silicon-PDMS membranes.

Surface characterization confirmed successful PEG attachment, with XPS revealing the appearance of C–O functional groups consistent with PEG functionalization, and water contact angle measurements indicating increased hydrophilicity. Oxygen flux tests demonstrated a reduction in gas transfer following PEG coating, likely due to the formation of a hydrated surface layer that adds a diffusion barrier [30,31]. To assess potential hemocompatibility benefits, we performed protein adsorption assays and found significantly reduced nonspecific protein binding on PEG-coated membranes compared to unmodified membranes. Since protein adsorption is an early initiator of thrombus formation, particularly on hydrophobic materials, this observation suggests improved compatibility with blood [32,33]. These findings underscore the importance of balancing surface modification strategies with the need to maintain adequate gas exchange in our membranes.

Surface modifications such as PEG offer biocompatibility benefits but can introduce trade-offs with gas exchange efficiency, a challenge also observed with other coatings like titanium dioxide, PEG acrylate, and heparin [25,34,35]. Consistent with these reports, we found that PEG functionalization reduced oxygen flux across multiple fluid flow rates and gas pressures, suggesting that this coating would impact oxygen transfer at scale, which is a critical consideration in designing a clinical-scale silicon membrane oxygenator. This reduction is likely due to surface wetting, where the hydrated PEG layer imposes a diffusion barrier at the gas–liquid interface [30,31]. While increasing membrane surface area could compensate for reduced flux, doing so would increase the blood-contacting area and potentially elevate thrombosis risk. Reducing the grafting density by lowering the initial PEG concentration (<0.5%) may create a thinner hydration layer and improve oxygen flux. Additional optimization strategies could include using lower molecular weight PEG, exploring alternative antifouling coatings, or accommodating for the reduced flux by increasing the membrane surface area. At the same time, PEG modification remains highly effective for mitigating protein adsorption on engineered surfaces, with significant reductions in HSA adsorption across multiple concentrations and surface areas. Previous studies by Kovach et al. and Gökaltun et al. similarly demonstrated that PEG-modified PDMS significantly reduces protein fouling [17,26]. Our results highlight the impact of PEG coatings on the oxygenator properties of silicon membranes, emphasizing the need to balance biocompatibility with gas transfer performance.

## 5. Conclusions

This work presents a systematic approach to advancing silicon membranes with PEG surface modifications for oxygenators by assessing key oxygenator properties. PEG coatings maintained hydrophilicity over 14 days and reduced protein fouling by up to 48%, indicating their promise for enhancing biocompatibility. Future work should explore the impact of variations in PEG layer density and PEG molecular weights on gas transfer and protein fouling. Further studies could examine platelet-poor-plasma or other blood proteins, such as fibrinogen or beta-globulin, under dynamic and in vivo conditions to evaluate long-term stability and clinical feasibility.

## Figures and Tables

**Figure 1 micromachines-16-01383-f001:**
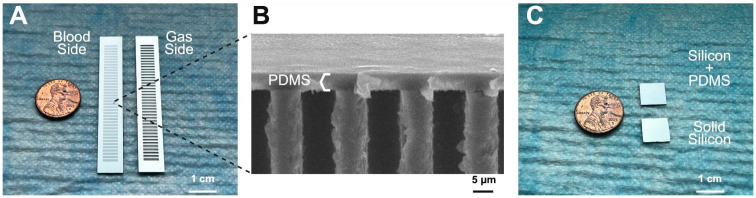
Silicon membranes used for PEG evaluation. (**A**) Composite membranes were used for gas exchange testing. The blood side was coated with the 5 μm PDMS and the oxygen side has open windows to allow for gas exchange. (**B**) Cross-sectional scanning electron microscopy (SEM) image of a composite membrane showing the PDMS bonded to the patterned silicon. (**C**) Solid silicon chips (1 cm × 1 cm) were used to mount the PDMS films for water contact angle, XPS, and protein adsorption measurements.

**Figure 2 micromachines-16-01383-f002:**
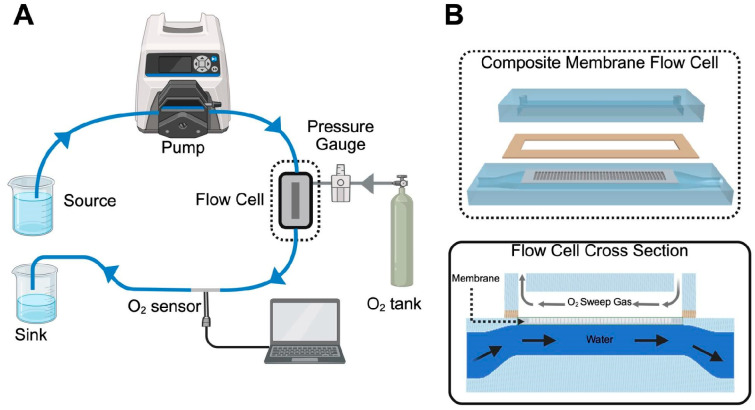
Oxygen flux circuit schematic. (**A**) Deoxygenated water is pumped from a peristaltic pump with L/S 16 tubing to a flow cell containing a membrane. Pressurized oxygen is also introduced to the flow cell allowing for gas exchange through the membrane. A NeoFoxGT O_2_ sensor measures dissolved oxygen content after the flow cell. (**B**) An exploded view of the composite membrane flow cell. The membrane lays in a recessed seat that is sealed with a gasket. The membrane separates the gas channel and water channel but allows for gas to permeate into the water. A cross-sectional view below shows the membrane separating the two channels. Water flows countercurrent to the oxygen.

**Figure 3 micromachines-16-01383-f003:**
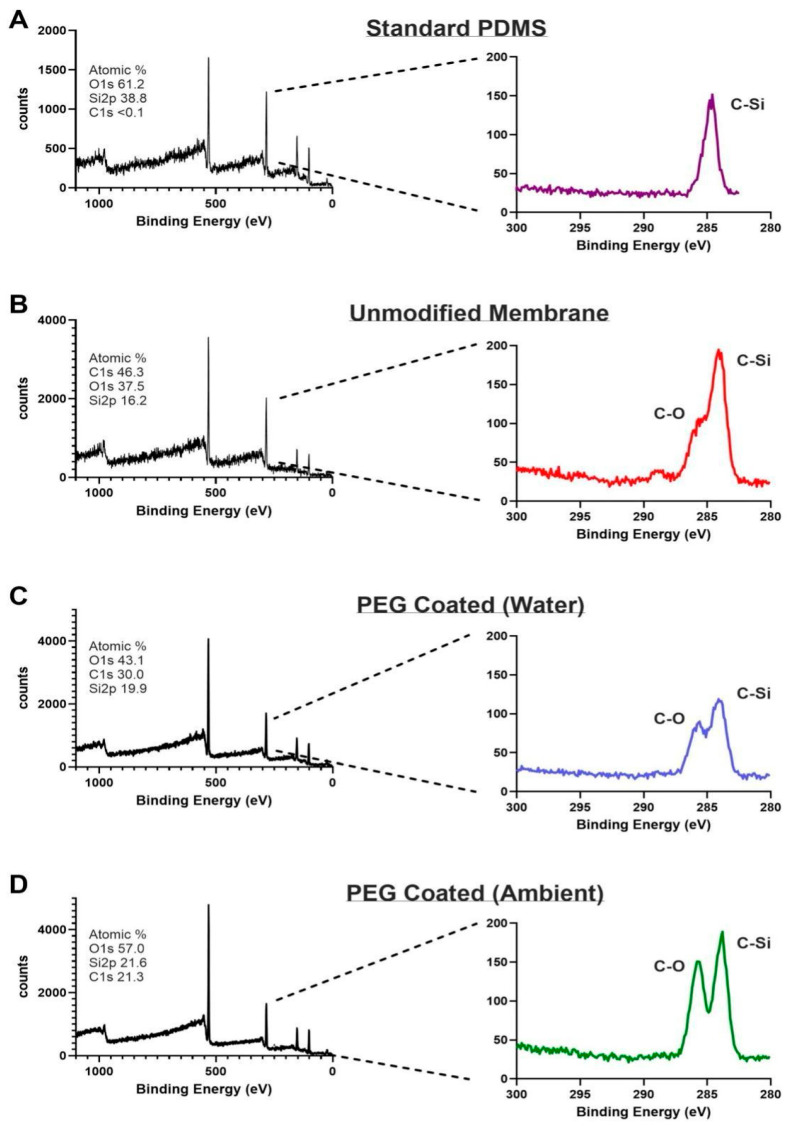
XPS spectra (left column) and high-resolution scans of C1s (right column). (**A**) Standard PDMS showing peak C-Si bond. (**B**) Unmodified membrane showing peak C-Si bonding as well as smaller C-O bond attributed to post-plasma treatment effects. (**C**) PEG coated membrane immersed in water exhibits both C-Si and C-O peaks confirming PEG modification. (**D**) PEG coated membranes stored in air similarly exhibit peaks for C-Si and C-O bonds.

**Figure 4 micromachines-16-01383-f004:**
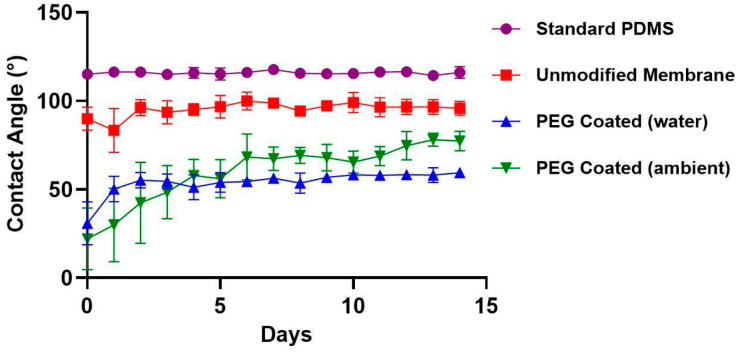
Water contact angle over two weeks of standard PDMS (●), unmodified membranes (■), PEG coated (water) (▲) and PEG coated (ambient) (▼). All samples were stored at room temperature in individual wells. PEG coated membranes stored in ambient air and DI water demonstrated lower contact angles compared to standard PDMS and unmodified membranes. Over the first 4 days, PEG coated membranes experienced some hydrophobic recovery due to oxygen plasma treatment effects.

**Figure 5 micromachines-16-01383-f005:**
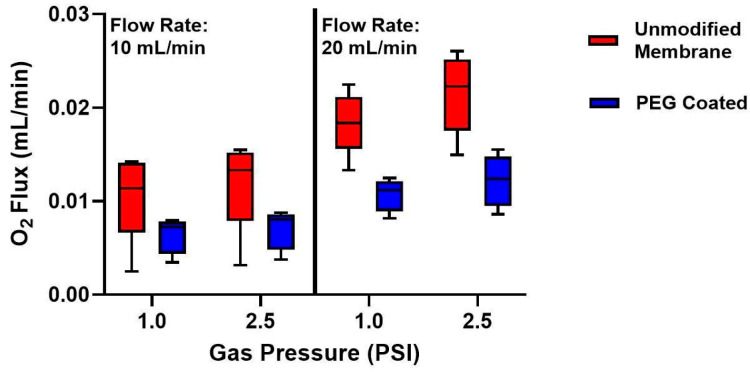
Oxygen flux measurements of composite membranes before and after PEG coating. Left: Oxygen flux at a fluid flow rate of 10 mL/min for 1.0 and 2.5 PSI oxygen gas pressure. There was a decrease in O_2_ flux from the unmodified membrane to the PEG-coated membrane at both gas pressures. Right: Oxygen flux at a fluid flow rate of 20 mL/min for 1 and 2.5 PSI oxygen gas pressure. At 20 mL/min, the overall flux was higher in both membrane groups, however a decrease in oxygen transfer from the unmodified membranes to the PEG-coated membranes again. (*n* = 5).

**Figure 6 micromachines-16-01383-f006:**
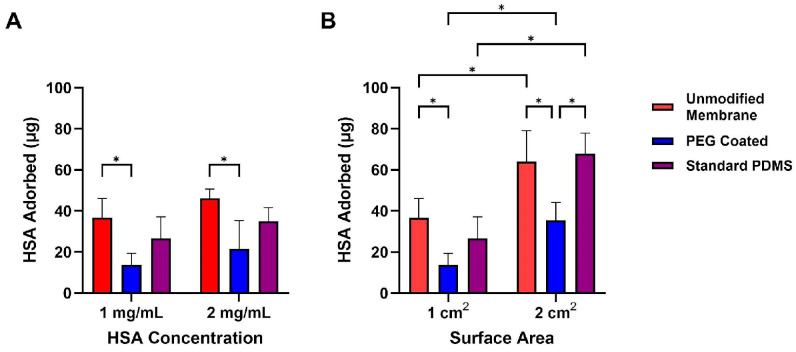
Protein adsorption of Human Serum Albumin onto composite membranes and standard PDMS samples after exposure to HSA for 1.5 h. (**A**) Comparison of HSA onto samples at 1 mg/mL and 2 mg/mL HSA concentration solutions. PEG-coated membranes were significantly lower in HSA adsorption compared to unmodified membranes. (**B**) HSA adsorption comparison of 1× and 2× surface areas of each group. At 2× surface area, PEG coated membranes demonstrated significantly lower protein adsorption compared to unmodified membranes and standard PDMS. Data are mean ± std. dev (*n* = 3). * *p* < 0.05.

## Data Availability

The original contributions presented in this study are included in the article. Further inquiries can be directed to the corresponding author.

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
