# Peer review of "Characterization of PEG-Modified Composite Membranes for Microfluidic Oxygenator Applications"

_micromachines, 2025, doi:10.3390/mi16121383_

Round 1
Reviewer 1 Report
Comments and Suggestions for Authors
1. While the study clearly shows that PEG modification significantly reduces protein adsorption, it also results in the reduction in oxygen flux. The authors are encouraged to elaborate on possible optimization strategies.
2. Protein adsorption experiments used only HSA as a model protein; it does not represent the full range of blood–material interactions. The authors should acknowledge this limitation and suggest broader future testing with multiple proteins or plasma.
Comments on the Quality of English LanguageThe English is generally clear and understandable, but some sentences are overly long. Minor editing for conciseness and clarity is recommended.
Author Response
Reviewer 1
Comment 1:While the study clearly shows that PEG modification significantly reduces protein adsorption, it also results in the reduction in oxygen flux. The authors are encouraged to elaborate on possible optimization strategies.
Response 1: Thank you for kind words regarding the study. To further elaborate on optimization strategies, we have added some additional text in the discussion page 11, lines 350-354. “PEG has previously been applied to flexible PDMS microfluidics and thin film composites such as polyamide [17], [18]. However, the application of PEG to thin film (<10 um) PDMS has yet to be studied.”
Comment 2: Protein adsorption experiments used only HSA as a model protein; it does not represent the full range of blood–material interactions. The authors should acknowledge this limitation and suggest broader future testing with multiple proteins or plasma.
Response 2: We completely agree that HSA does not fully represent the variety of proteins in blood. We acknowledge this limitation in the text, page 12 lines 366-369. “Future work should explore the impact of variations in PEG layer density and PEG molecular weights on gas transfer and protein fouling. Further studies could examine platelet-poor-plasma or other blood proteins, such as fibrinogen or beta-globulin"
Please see updated manuscript if necessary and thank you for the review.
Reviewer 2 Report
Comments and Suggestions for Authors
This paper proposed a microfluidic based Extracorporeal membrane oxygenation (ECMO). The main work in their manuscript is the surface modification by applying the PEG Coating. Their platform is well designed and the fabrication is clearly. The quantitate results are clear for the readers to understand the results. All in all, it is clear and this manuscript should be accepted after a minor revision.
- Some description in the main content need to be checked to match with the figures. For example in Introduction, line 40. you mention "Figure 1" I think it is not correct.
- The introduction should add more information to have a clear background and research gap.
- In your method section, you mention you make the 5um think PDMS layer. I observe you mention the mixture of hexane and PDMS. Therefore, what is the spin coated rpm? How you control whether it is uniform for the whole surface?
- Do you have any microscope images of membrane fluctuation when you inject the water in Fig.2?
- I think it is better to optimize the quality of the images. it is better to control the font of the words and number for all the images.
- It is better to add more discussion sentence in the results section to make the content more clear.
Author Response
Reviewer 2
Comment 1: This paper proposed a microfluidic based Extracorporeal membrane oxygenation (ECMO). The main work in their manuscript is the surface modification by applying the PEG Coating. Their platform is well designed and the fabrication is clearly. The quantitate results are clear for the readers to understand the results. All in all, it is clear and this manuscript should be accepted after a minor revision.
Response 1: Thank you for the review, suggestions, and kind words regarding the manuscript. We have addressed your comments and concerns and replied to them below.
Comment 2: Some description in the main content need to be checked to match with the figures. For example in Introduction, line 40. you mention "Figure 1" I think it is not correct.
Response 2: Thank you for catching this. We were referring to the graphical abstract, so we have struck this from the manuscript to reduce confusion.
Comment 3: The introduction should add more information to have a clear background and research gap.
Response 3: We understand the need to improve the introduction to make it clearer. To emphasize the research gap concern, we added text to address this on Page 3, lines 72-74. “PEG has previously been applied to flexible PDMS microfluidics and thin film composites such as polyamide [17], [18]. However, the application of PEG to thin film PDMS has yet to be studied."
- Z. Sanei, T. Ghanbari, and A. Sharif, “Polyethylene glycol-grafted graphene oxide nanosheets in tailoring the structure and reverse osmosis performance of thin film composite membrane,” Sci Rep, vol. 13, no. 1, p. 16940, Oct. 2023, doi: 10.1038/s41598-023-44129-z.
Comment 4: In your method section, you mention you make the 5um think PDMS layer. I observe you mention the mixture of hexane and PDMS. Therefore, what is the spin coated rpm? How you control whether it is uniform for the whole surface?
Response 4: The RPM used in our protocol is 5000 rpm for 5 minutes. We have previously published on this protocol in citation 17. To provide more clarity, we added the RPM and time to the material and methods for all the stepspage 4:
- Handle PDMS: lines 88-89 “at 500 rotations-per-minute for 20 seconds.”
- PVA layer: line 94, “at 1000 rotations per minute for 30 seconds and curing at 60°C”
- Thin PDMS: lines 96-98, “at 5000 rotations-per-minute for 5 minutes to a highly uniform and controlled thickness of approximately 5 μm, and subsequently cured”
- D. G. Blauvelt et al., “A silicon membrane microfluidic oxygenator for use as an artificial placenta with minimal anticoagulation,” Bioengineering & Transla Med, July 2025, doi: 10.1002/btm2.70037.
Regarding the control of uniformity, spin coating is known and well-studied to provide a uniform surface from the strong centrifugal forces. Additionally, we cut the PDMS from the handle wafer from the center to remove any variation that could occur near the edges. We do acknowledge that there could be slight variations between films; however, we have yet to observe any significant variation in our testing.
Comment 5: Do you have any microscope images of membrane fluctuation when you inject the water in Fig.2?
Response 5: Thank you for the interesting question. At these relatively low flow rates, we do not expect any deflection or fluctuation in the rigid silicon backbone. We have previously published showing the deflection of the PDMS component at 30 PSI is less than 1 um so at these lower pressures (<2 PSI), we do not anticipate any fluctuation of the membrane itself.
- Blauvelt, D.G., Chui, B.W., Higgins, N.C. et al. Silicon membranes for extracorporeal life support: a comparison of design and fabrication methodologies. Biomed Microdevices 25, 2 (2023). https://doi.org/10.1007/s10544-022-00639-7
Comment 6:I think it is better to optimize the quality of the images. it is better to control the font of the words and number for all the images.
Response 6: Thank you for noticing quality concerns of the images. To make improvements, we have reviewed and updated the fonts and sizing of words/numbers in the images. The graphical abstract and Figures 1,2 have been updated to reflect these changes.
Comment 7: It is better to add more discussion sentence in the results section to make the content more clear.
Response 7: We completely understand the need for clarity. To address this we added a brief discussion to each results section.
- XPS: lines 233-235, “The enhanced C–O signal in the PEG-coated samples confirms that the spin-coating and plasma treatment method can lead to successful surface functionalization of the thin-film PDMS.”
- Contact Angle: lines 260-261, “The data suggests that storage in water will extend shelf-life of the PEG-coated membranes.”
- Oxygen Flux: lines 282-284, “ While there is indeed a decrease in oxygen flux with PEG coatings, this effect can be, at least partially, counteracted by operating at increased fluid flow rates.
- Protein Adsorption: lines 308-309, “As such, PEG is indeed promising as a coating to enhance hemocompatibility.”
Please see the updated manuscript attachment if necessary.